# Examining the Relationship between Land Use/Land Cover (LULC) and Land Surface Temperature (LST) Using Explainable Artificial Intelligence (XAI) Models: A Case Study of Seoul, South Korea

**DOI:** 10.3390/ijerph192315926

**Published:** 2022-11-29

**Authors:** Minjun Kim, Dongbeom Kim, Geunhan Kim

**Affiliations:** 1Department of Environmental Planning, Korea Environment Institute, Sejong 30147, Republic of Korea; 2Technical Research Institute NEGGA Co., Ltd., Seoul 07220, Republic of Korea

**Keywords:** land-use/land-cover (LULC), land surface temperature (LST), explainable artificial intelligence (XAI), Sharpley additive explanations (SHAP), remote sensing (RS)

## Abstract

Understanding the relationship between land use/land cover (LULC) and land surface temperature (LST) has long been an area of interest in urban and environmental study fields. To examine this, existing studies have utilized both white-box and black-box approaches, including regression, decision tree, and artificial intelligence models. To overcome the limitations of previous models, this study adopted the explainable artificial intelligence (XAI) approach in examining the relationships between LULC and LST. By integrating the XGBoost and SHAP model, we developed the LST prediction model in Seoul and estimated the LST reduction effects after specific LULC changes. Results showed that the prediction accuracy of LST was maximized when landscape, topographic, and LULC features within a 150 m buffer radius were adopted as independent variables. Specifically, the existence of surrounding built-up and vegetation areas were found to be the most influencing factors in explaining LST. In this study, after the LULC changes from expressway to green areas, approximately 1.5 °C of decreasing LST was predicted. The findings of our study can be utilized for assessing and monitoring the thermal environmental impact of urban planning and projects. Also, this study can contribute to determining the priorities of different policy measures for improving the thermal environment.

## 1. Introduction

Due to rapid urban growth over the last few decades, soils and vegetation have been converted into impervious materials such as concrete, asphalt, and buildings, which induce changes in the city’s biophysical environment and energy processes [1]. These shifts also affect the degree of solar radiation absorption, albedo, evaporation rate, heat transfer to soil, heat storage, wind turbulence, and can significantly alter atmospheric conditions near the urban surfaces [2].

The urban heat island (UHI) phenomenon refers to how the air (or surface) temperature of urban areas is warmer than their rural surroundings [3]. Because of its negative impact on the ecology and viability of cities, UHI has become a major research focus in various interrelated fields, including urban climatology, urban ecology, urban planning, and urban geography [4]. In particular, an increase in land surface temperature (LST) induced by UHI can impair the composition and distribution of species by increasing the length of the growing season [5,6], and it can worsen air quality [7,8] and increase health risks [9].

LST is a key variable for understanding the impact of land use/land cover (LULC) changes due to urbanization [10]. Accordingly, many researchers have studied the relationship between LULC and LST using various methods. In early studies, researchers utilized multiple regression models to compare LST for each LULC type [2,11]. Using satellite images such as Landsat, they showed that LST is highly associated with adjacent green space and built-up areas [12].

However, as many pointed out that the relationship between LULC and LST is not linear, recent studies have adopted more advanced methods to analyze them. Weng et al. [13] combined the maximum likelihood method and decision tree algorithm in analyzing the relationship between LST, LULC, and NDVI. Rana and Suryanarayana [14] analyzed the contribution of various LULC types on LST by comparing four machine learning models (K-Nearest Neighbor, Artificial Neural Network, Random Tree, and Support Vector Machine). Bakar et al. [15] used a support vector machine and maximum likelihood method to investigate the relationship between LST and LULC. Gage and Cooper [16] evaluated the relative importance of LULC, NDVI, and vertical structure for LST patterns through random forest modeling. Kafy et al. [17] used the Multi-Layer Perceptron-Markov Chain and Artificial Neural Network (ANN) methods to simulate LST and LULC. Bozorgi et al. [18] used an artificial neural network (ANN) algorithm to explore the relationship between LST and green LULC spatial patterns.

While those machine learning (ML) and artificial intelligence (AI) models have improved the overall prediction accuracy of LST, the black-box nature of these techniques has inevitably lowered the interpretability [19]. To overcome these methodological limitations, the explainable artificial intelligence (XAI) approach has been recently highlighted. XAI is a methodology that provides interpretation so that humans can understand the results predicted by the machine learning algorithm [20]. In this regard, XAI models require an additional explainable algorithm, such as Shapley additive explanations, to interpret how the model achieved a specific result [21].

Moreover, a majority of previous studies have focused on examining the effects of certain LULC types (parks, rivers, etc.) or landmarks (buildings, factories, etc.) on increasing (or decreasing) LST [22,23]. To consider various factors that influence LST, however, it is necessary to analyze the associations between LST within a whole region and various LULC types nearby. Furthermore, it is essential to figure out the appropriate range of nearby LULC features for LST prediction, which is not yet fully addressed in the existing literature. This can be achieved by developing multiple prediction models with different buffer distances from each of the LST points.

The research question of this study is which are the surrounding LULC types that affect LST, and what is the extent of these effects. Furthermore, our main concern is whether the advanced AI techniques provide accurate LST prediction that can be applied to the urban and environmental planning sector. To this end, this study aims at examining the impact of various LULC types on LST using XAI models in Seoul, Korea. By integrating the XGBoost and SHAP model, we developed the LST model for several buffer distances and predicted the LST reduction effects after LULC changes. More specifically, an expressway that would be undergrounded was chosen and assumed to be converted into green park areas. By utilizing the developed LST model, we predicted the LST reduction effects on the regions surrounding the expressway. The findings of our study may contribute not only to assessing the thermal and environmental impact of urban planning, but also to determining the priorities of different policy measures in improving the thermal environment.

This paper is organized as follows. Chapter 2 describes the materials and methods of research. The dependent variable was the LST in Seoul, and landscape, topography, and LULC were independent variables in the LST model. In addition, the pre-processing steps, mathematical formulas for building the proposed LST model, and post-processing steps are described. Chapter 3 provides training, validation, and testing results for the LST predictive model. Finally, Chapters 4 and 5 discuss the results of this experiment, and present the conclusion and future tasks.

## 2. Materials and Methods

### 2.1. Study Area

Seoul is the capital city of Korea, with an area of 606 km^2^ and a population of 9.8 million, and it is one of the most densely populated cities in the world. The city has a high proportion of elderly population who are vulnerable to heat waves, and their proportion of the population is expected to continuously increase [24].

For several reasons, Seoul is one of the most suitable cities for examining the relationships between LST and LULC. First, Seoul has a highly mixed pattern of LULC (Figure 1a). A majority of the city’s land is currently covered by urbanized areas with residential and commercial use, while mountainous and agricultural areas are distributed on the outskirts of the city [25]. The surface of Seoul is generally flat and plain, and the Han River runs through the city center.

Second, Seoul is located in the center of Northeast Asia, and thus has a representative continental climate where it is cold and dry in winter but hot and humid in summer periods [26]. Due to densely distributed built-up areas within the city, Seoul has experienced a severe UHI phenomenon every summer, where the LST of urbanized areas is 10 to 13 °C higher than surrounding rural areas [27]. It provides an environment in which the association between LST and LULC can be analyzed more variously from temporal and spatial perspectives.

In order to examine the practical applicability of using the model built through these experiments, the Gyeongbu Expressway underground project site (6.8 km) was selected as the test subject for Seoul when the area, a large-scale transportation facility, was transformed into a green space (Figure 1b). We tried to analyze the LST-reduction effect by comparing and analyzing the existing LST. The area is being promoted for the purpose of resolving extreme traffic congestion and securing self-regulating green space in the ground-level space, and the feasibility of the current plan is being reviewed. By understanding the relationship between the LST and various factors constituting the city, it is judged that it will be very helpful in urban planning and policies to prevent the urban heat island in urban space.

### 2.2. Data

To analyze LST in Seoul, we constructed landscape, topographic, and LULC feature data for 2019. Table 1 summarizes the dependent and independent variables used in the study.

#### 2.2.1. Dependent Variable

In this study, the dependent variable was LST in Seoul, which was derived from a Landsat 8 satellite image of 13 June 2019 (Figure 2a). Landsat 8 is currently operated by the United States Geological Survey (USGS) and is accessible using Earth Explorer (available online: http://earthexplorer.usgs.gov/ (accessed on 1 June 2022)).

To obtain LST from the Landsat 8 image, we utilized band 10 from the thermal infrared sensor (TIRS) and band 3, 4 from the operational land imager (OLI) sensor, as the USGS suggested [28]. First, top of atmospheric radiance (TOA, $L_{\lambda }$) is calculated as below:(1)$$L_{\lambda }=M_{L}Q_{cal}+A_{L}$$
where $Q_{cal}$ denotes pixel values of band 10 (DN), $M_{L}$ (3.34 × 14^−4^) and $A_{L}$ (0.1) denote band-specific multiplicative and additive rescaling factor.

Using TOA, we then calculated at-satellite brightness temperature (T) as:(2)$$T=\frac{K_{2}}{ln\left(\frac{K_{1}}{L_{\lambda }}+1\right)}$$
where $K_{1}$ (774.89) and $K_{2}$ (1321.08) indicate band-specific thermal conversion constants. To convert values from Kelvin (*K*) to Celsius degree (°C), we subtracted 273.15 from the brightness temperature (T).

To calculate LST using brightness temperature (T), the surface emissivity needs to be obtained first. This study adopted the simplified normalized difference vegetation index (*NDVI*) approach to estimate it [29]. Using *NDVI*, the proportion of vegetation (*PV*) and the surface emissivity (ε) are calculated as below:(3)$$PV={\left(\frac{NDVI-NDVI_{min}}{NDVI_{max}-NDVI_{min}}\right)}^{2}$$
(4)ε=0.004×PV+0.986
where NDVI is the actual index while $NDVI_{max}$ and $NDVI_{min}$ are the NDVI ranges used between soil and vegetated area.

Lastly, using the brightness temperature (T) and surface emissivity (ε), *LST* is calculated as follows:(5)$$LST=\frac{T}{1+\frac{0.00115\times T}{1.4388}\times ln\left(\varepsilon \right)}$$

#### 2.2.2. Independent Variables

Independent variables of the study consist of (1) landscape, (2) topography, and (3) LULC features. All features were derived from remotely sensed data and have a 10 m spatial resolution.

First, landscape features include normalized difference built-up index (*NDBI*), normalized difference water index (*NDWI*), and green normalized difference vegetation index (*GNDVI*). These indices were derived from a Sentinel-2 satellite image of 23 May 2019, which was provided by the European Space Agency (ESA). Different from urban and vegetation LULC types, the NDBI and GNDVI indices reflect the building density and the vitality of vegetation, respectively [30].

To calculate these, we utilized band 3 (*Green*), band 8 (*NIR*), and band 11 (*SWIR*) of Sentinel-2, as below [31]:(6)$$NDBI=\frac{\left(SWIR-NIR\right)}{\left(SWIR+NIR\right)}$$
(7)$$NDWI=\frac{\left(NIR-SWIR\right)}{\left(NIR+SWIR\right)}$$
(8)$$GNDVI=\frac{\left(NIR-Green\right)}{\left(NIR+Green\right)}$$

Second, this study utilized elevation and slope as topographic features. To calculate those, a digital elevation model (DEM) dataset was obtained from the National Spatial Data Infrastructure portal (NSDI, available online: http://www.nsdi.go.kr/ (accessed on 1 August 2022)). Using the Surface tool in ArcGIS 10.1 software, the10 m resolution elevation and slope for Seoul were calculated (Figure 2e,f).

Third, for land-cover features, we used a 10 m resolution sub-divided land-cover map of 2019, which was provided by the Environmental Geographic Information Service (EGIS, available online: http://egis.me.go.kr/ (accessed on 1 August 2022)). In this study, land cover in Seoul was classified into 23 types including urbanized, agricultural, forest, grassland, wetland, bareland, and water area.

### 2.3. Methods

#### 2.3.1. Research Procedure

Figure 3 illustrates the study’s overall research procedure. To derive the LST model, we estimated landscape, topographic, and land-cover features within a 50 to 500 m buffer radius from each LST raster cell. The LST value is increased in 50 m increments from 50 to 500 m by performing a buffer centered on each pixel.

Then, this study combined the extreme gradient boosting (XGBoost) and Sharpley additive explanations (SHAP) to develop the LST model. For each dataset, 80% of samples were utilized to train the model, and 20% of them were left for validation. Among all the LST models that were developed from the XGBoost algorithm, we compared the overall accuracy including RMSE and pseudo R^2^ and chose the optimal LST model with a certain buffer radius through sensitivity analysis. Last, for the optimal LST model, this study calculated SHAP values for each independent variable and analyzed relative importance and direction of predicting LST.

Using the optimal LST model, this study predicted the effects of LULC change in Seoul on surrounding LST. As the case study area, we selected the expressway in the south-eastern part of the city, which will be undergrounded until 2025. While there are still no final plans for utilizing these areas, we assumed that the existing expressway infrastructures would be converted into green park areas and predicted the LST after the LULC changes.

#### 2.3.2. Extreme Gradient Boosting (XGBoost) Model

Boosting-based machine learning techniques combine a series of weak learners to build a strong learner [32]. Among these, gradient boosted decision trees (GBDT) are performed in a way that sequentially reduces errors in the tree model by reflecting residuals calculated from the previous one [33]. The XGBoost model is an algorithm developed based on GDBT, which improves computational speed and prevents overfitting problems [34]. Thanks to its high predictive precision and ability to deal with both classification and regression problems, the GBDT algorithm has been widely used in various disciplines [35]. This study utilized the “xgboost” package in R (ver. 1.5.2.1.) for the analysis, which was released in February 2022.

Figure 4 illustrates the overall process of the XGBoost model. In the model, decision tree *K*(X, $\theta_{k}$) is trained from the residuals of the previous tree *K*−1(*X*, $\theta_{k-1}$), and additively builds a more powerful learner until the loss function $f_{k}(X,$ $\theta_{k}$) is minimized [36].

To maximize the model’s prediction power while preventing over-fitting issues, there are several parameters that need to be designated in the XGBoost algorithm. Parameters include the number of iterations, maximum depth, the fraction of observations, and learning rate. For optimal hyper-parameter tuning, this study adopted fivefold cross-validation techniques.

#### 2.3.3. Shapley Additive Explanations (SHAP)

The Shapley additive explanation (SHAP) is one of the representative explainable artificial intelligence (XAI) methods that have been widely used to increase the interpretability of machine learning models [37]. The SHAP value quantifies the relative importance of each independent variable for the model outcome based on its marginal contribution [38]. For n features, the SHAP value $\phi_{i}$ assigned to each feature *i* is represented as below:(9)$$\phi_{i}=\sum_{S\in F\backslash \left\{i\right\}}\frac{\left|S\right|!\left(\left|F\right|-\left|S\right|-1\right)!}{\left|F\right|!}\left[f_{S\cup^{\;}\left\{i\right\}}\left(x_{S\cup^{\;}\left\{i\right\}}\right)-f_{S}\left(x_{S}\right)\right]$$
where F represents all the combinable features in *S*, and $f_{S\cup^{\;}\left\{i\right\}}\left(x_{S\cup^{\;}\left\{i\right\}}\right)-f_{S}\left(x_{S}\right)$ calculates the difference between the contributions when the feature *i* is used and when it is not used.

## 3. Results

### 3.1. Descriptive Statistics

Table 2 summarizes the descriptive statistics of dependent and independent variables used in the study. First, the average LST of Seoul was 27.35 °C, which ranges from 13.62 °C to 43.13 °C. Areas with relatively high LST were generally covered with urbanized area, while those with low LST were associated with green and water areas (Figure 2a).

The landscape features, including NDBI, NDWI, and GNDVI, show negative average values. The average NDBI, however, was higher than the other two indices, which implies that Seoul’s lands are more occupied by built-up areas compared to natural environment. Average elevation and slope in Seoul were 59.93 and 7.9, respectively, indicating that there are relatively gentle topographic characteristics throughout the city.

The proportion of each LULC type in Seoul is described in Figure 5. The majority of Seoul’s LULC is occupied by urban areas (60%), particularly for transportation (28%), residential (13%), and commercial areas (8%). Among the urban LULC, industrial area (0.18%) covers the smallest ratio. This suggests that urbanized areas within the city are being predominantly utilized for the citizens’ livelihood.

Meanwhile, forest (25%) and grassland (12%) have the largest proportion within the natural LULC in the city, followed by water (5%) and bareland area (5%). More specifically, most of the natural areas in Seoul consist of broadleaf forest (16%), artificial grassland (11%), and inland water (5%). However, there were a few agricultural (2%) and wetland areas (1%) throughout the city.

### 3.2. Model Results

#### 3.2.1. Hyper-Parameter Tuning

Table 3 summarizes the optimal hyper-parameter values chosen for the XGBoost model developed in this study. First, the number of iterations and maximum depth were 280 and 11, respectively. Among samples, 70% of them were randomly extracted for training the model, with a learning rate of 0.2. Based on the fivefold cross-validation, ‘colsample_bytree’, ‘alpha’, ‘lambda’, and ‘gamma’ values were tuned as 0.9, 1 × 10^−6^, 0.05, and 0, respectively, to prevent the overfitting issues of the XGBoost model.

#### 3.2.2. Sensitivity Test

Using the derived hyper-parameters, we developed 10 XGBoost models with different buffer distances from each LST raster cell. Figure 6 illustrates the sensitivity of prediction accuracy with respect to a buffer radius from 50 m to 500 m.

For a 50 m buffer, root mean square error (RMSE) and R-squared (R^2^) values of the model were 1.32 and 0.882, respectively. The prediction accuracy was sharply increased until 150 m of buffer radius, where RMSE and R^2^ values reached 1.09 and 0.918. After 150 m, however, the RMSE of the LST model steadily increased to 1.35, while R^2^ decreased to 0.876 until 500 m of buffer radius. As a result, the optimal buffer distance for the LST model was chosen as 150 m in this study.

#### 3.2.3. Factor Importance

Based on the sensitivity test results, this study estimated SHAP values of the LST model for 150 m buffer radius (Figure 7). Figure 7a shows the relative importance of influencing factors on LST, where the x-axis indicates the absolute SHAP value for each variable. As the length of the absolute SHAP value increases, the amount of how much a single feature affected the prediction also increases. Figure 7b represents the original SHAP value, where a high feature value (red color) indicates the direction of influencing factors on LST. When the SHAP value of a certain feature is positively distributed, it implies that a feature positively affects the prediction.

First, NDBI and GNDVI were found to be the most important predictors of LST. More specifically, the LST of a certain raster cell increases as the surrounding region is more urbanized and less vegetated. NDWI also tended to be negatively associated with LST. This corresponds with previous studies’ findings that the existence of nearby impervious and natural environments have opposite effects on LST [39].

With regards to topographic features, adjacent DEM and Slope showed relatively high association with LST, but their direction of effects were not clear. One of the possible reasons for this result is that Seoul has relatively gentle topographic features, and thus there is no significant difference in LST depending on them [40]. For LULC features, the existence of inland water (710) showed strong negative associations with LST. It is not surprising that the average temperature of riverside areas is relatively low [41]. On the other hand, surrounding built-up areas, including commercial (130), residential (110), and transportation areas (150), were found to be positively associated with LST, which is in line with findings from previous works [42]. It is notable that the existence of commercial and residential areas is more important for determining LST compared to transportation areas. Regarding natural LULC, artificial bareland (620), broadleaf forest (310), and artificial grassland (420) were some of the most influencing factors in explaining LST.

#### 3.2.4. LST Prediction

Using the optimal LST model, we predicted the LST before and after the expressway underground project in Seoul. For the comparison, the study assumed that all transportation areas (150) along the expressway would be converted to artificial grassland (420). To examine the effect of LULC changes on the LST itself, we entered the average values of landscape (NDBI, NDWI, GNDVI) and topographic (elevation, slope) features of artificial grassland (420) in Seoul when predicting the LST.

Figure 8 and Figure 9 show the distribution of LST before and after the expressway underground project with several buffer distances from the expressway. Prior to the project (Figure 8a,b), the average LST along the expressway was 27.51 °C, which ranges from 22.50 °C to 30.97 °C. After the greening (Figure 8c,d), however, the average LST within the expressway was decreased by 0.39 °C, and the largest decrease was about 1.59 °C. This LST reduction effect was found to be more pronounced near the expressway intersection, where the proportion of impervious areas is generally high [43].

As the distance from the expressway increases, the LST reduction effect was steadily weakened. It is noteworthy that there is little effect on reducing LST by greening expressway from a radius of 150 m or more. Similarly, the maximum LST reduction was 1.63 °C, and it was sharply decreased after 150 m away from the expressway. This trend corresponds with the LST prediction models the study developed. In addition, our results suggest that when the built-up area is converted to green space, the LST reduction effect can be brought to not only the area itself but also the neighboring area.

## 4. Discussion

Compared to previous studies, our study is novel in both methodological and theoretical perspectives. First, the study adopted an XAI approach in examining the impact of LULC changes on LST. In early studies, the prediction accuracy of the LST model was relatively low since the associations between influencing factors and LST had been understood to be linear [44]. To capture the non-linear properties of LST modeling, recent studies have adopted several advanced techniques including machine learning (ML) and artificial intelligence (AI) models [18]. As a result, the overall accuracy of LST prediction has been improved, but their black-box nature has inevitably lowered the interpretability of models [45].

To overcome these methodological limitations, we utilized the XAI model by integrating XGBoost and SHAP models. This enabled our study to sufficiently interpret the associations between several remotely sensed features and LST, while guaranteeing the relatively high prediction accuracy. Second, this study developed several LST models and implemented the sensitivity test to derive the optimal buffer radius for landscape, topographic, and LULC features from a certain LST raster cell. The findings of this study showed that the overall accuracy of models peaks at a 150 m buffer radius, and then steadily decreased with longer distances. While there are a number of recent studies that have examined the LST reduction effect at several distances from a specific LULC type, including a park or a river [23,46], only a few studies have examined the sensitivity of model accuracy by differing distances from LST.

Third, our study’s results showed that NDBI and GNDVI are some of the most influencing factors in predicting LST. It is noteworthy that qualitative rather than quantitative aspects of the surrounding environment may contribute more to lowering LST. In addition, by using the developed model, we predicted the LST reduction effects by converting LULC types. Results showed that replacing built-up areas with vegetation can significantly decrease the nearby LST.

Based on this study’s findings, we propose several policy implications for planning and managing the urban thermal environment. First, current environmental planning promotes greening projects and low-density development throughout the city to reduce thermal environmental issues such as UHI. However, the thermal environment of the city is spatially heterogeneous, and thus it is necessary to consider regional characteristics in establishing strategies. To this end, practitioners in the urban and environmental planning sector should evaluate the spatial extent of various factors that affect the air (or surface) temperature within the city. The optimal extent of thermal impact may depend on the city’s built-environment and climate characteristics.

Second, the XAI-based LST prediction model can be utilized for assessing and monitoring the thermal environmental impact of urban planning and projects. More specifically, planners can construct LST prediction maps and can evaluate the effects of multiple urban development (or greening) projects on thermal environments. Moreover, with the high reliability of the model and the derived relative importance between variables, this method can contribute to determine the priorities of different policy measures. Third, urban and environmental planners should take several strategic measures to preserve green areas within the city that are found to be effective in reducing LST. In particular, for the areas with relatively high LST, including commercial, road, and parking areas, it is necessary to establish measures to secure or to link with adjacent green areas. Otherwise, the LST of urban areas can be lowered by constructing narrow green areas in between the roads and buildings.

Finally, this study suggests that both quantitative and qualitative improvement is necessary to increase the city’s thermal comfort. While the current environmental planning has focused on securing a sufficient number of LULC types (park, river, etc.) that reduce thermal environmental issues, the quality of those areas (shape, vitality, etc.) has not been fully considered [47]. In order to improve the thermal environment in the city, it is necessary to evaluate the surrounding environment of vulnerable areas and establish site-specific strategies with both quantitative and qualitative perspectives.

## 5. Conclusions

Examining the impact of LULC changes on LST has long been an area of interest in urban study and remote sensing fields. By integrating the XGBoost and SHAP model, this study developed the LST model for several buffer distances and predicted the LST reduction effects after LULC changes. The findings of this study showed that the prediction accuracy of LST was maximized when landscape, topographic, and LULC features within a 150 m buffer radius were adopted as independent variables. As a result, the existence of surrounding built-up and vegetation areas were found to be the most influencing factors for explaining LST. Moreover, LULC changes from expressway to green park areas were predicted as more than 1.5 °C of decreasing LST, which supports the reliability of the model.

The findings of this study contribute to the existing literature as it applied advanced XAI models in examining the effects of LULC change on LST. Since dependent and independent variables used in this study are easily accessible, the same analysis can be performed for other regions. Based on its generality, the research process of this study could be widely applied as a preliminary tool for establishing strategies of thermal environment planning.

Despite this study’s contribution, there are still several limitations that need to be improved in future studies. First, this study utilized a single remotely sensed image to analyze the relationships between LST and LULC. To capture the diurnal and seasonal variations of LST, it is necessary to acquire a sufficient number of satellite images in further research [48]. Second, while it is obvious that LST is largely affected by the albedo of land surface [49], the independent variables used in the study could not reflect these characteristics in the model. When future studies include building (or road) materials and reflectivity in modeling LST, it is expected that more in-depth urban planning and design measures can be derived [50]. Finally, there have been a number of XAI models, such as LightGBM and Catboost, developed and applied in recent studies [51,52]. Predicting LST using various XAI models and comparing the results would contribute to improving methodological aspects of thermal environment modeling.

## Figures and Tables

**Figure 1 ijerph-19-15926-f001:**
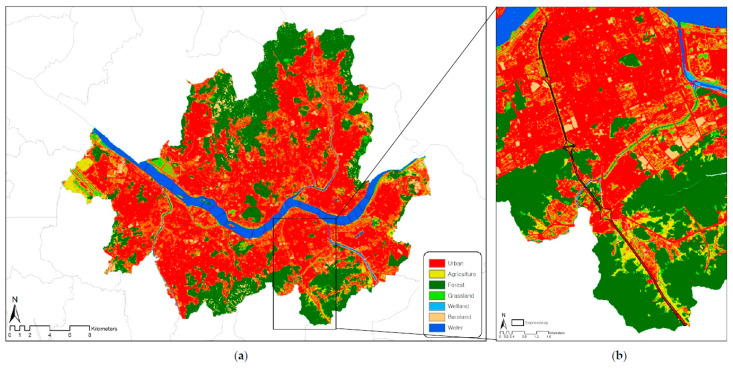
Study area (**a**) Seoul city and (**b**) Gyeongbu Expressway.

**Figure 2 ijerph-19-15926-f002:**
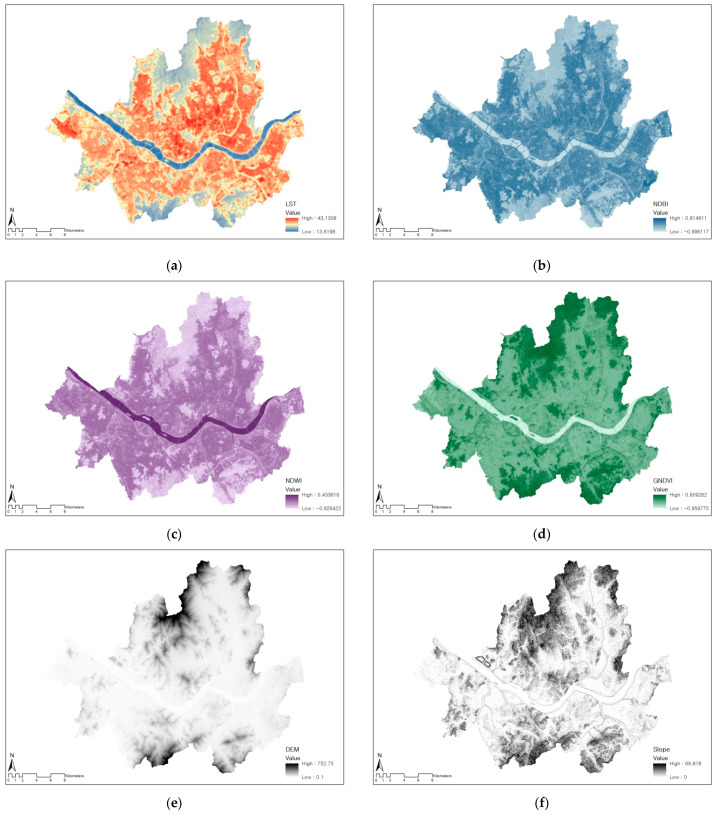
Variables used in the study. (**a**) LST, (**b**) NDBI, (**c**) NDWI, (**d**) GNDVI, (**e**) elevation, (**f**) slope.

**Figure 3 ijerph-19-15926-f003:**
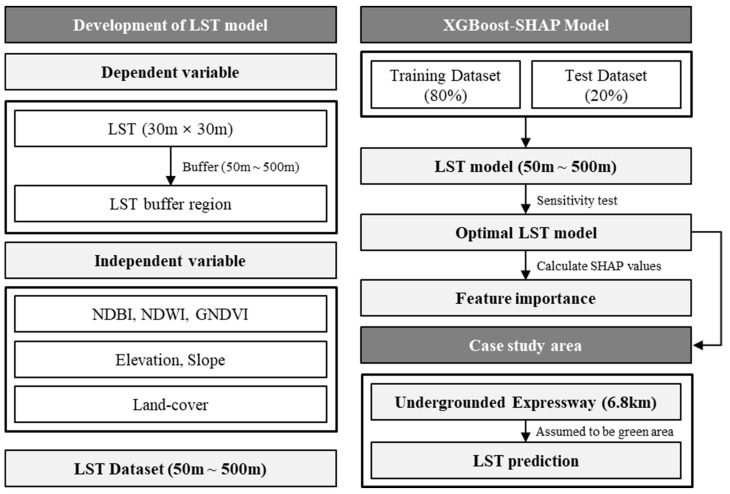
Research process.

**Figure 4 ijerph-19-15926-f004:**
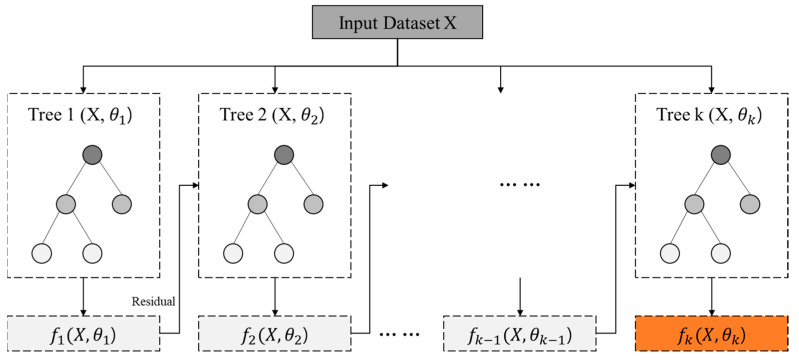
XGBoost model process.

**Figure 5 ijerph-19-15926-f005:**
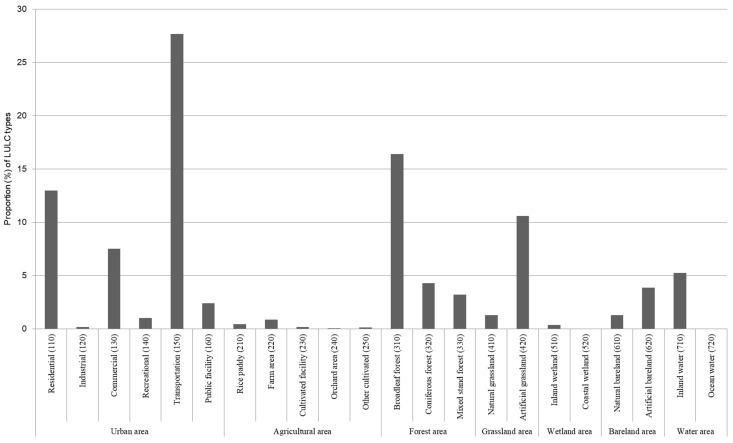
Proportion of LULC types in Seoul.

**Figure 6 ijerph-19-15926-f006:**
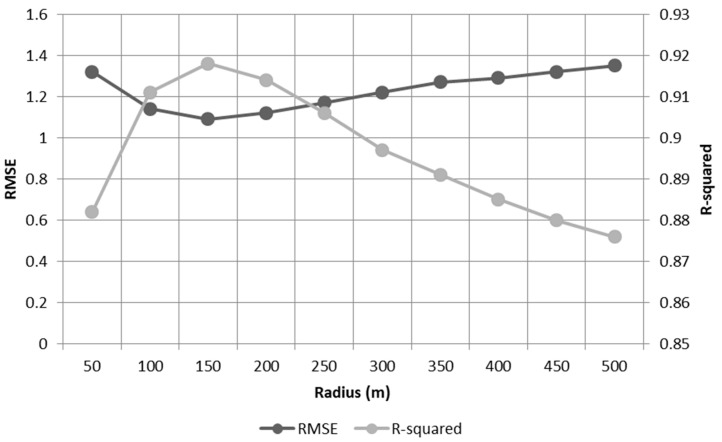
Sensitivity of prediction accuracy.

**Figure 7 ijerph-19-15926-f007:**
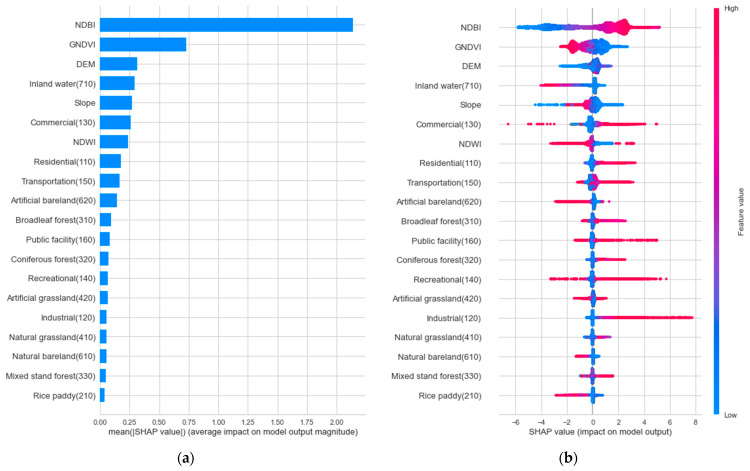
(**a**) Relative importance and (**b**) direction of independent variables.

**Figure 8 ijerph-19-15926-f008:**
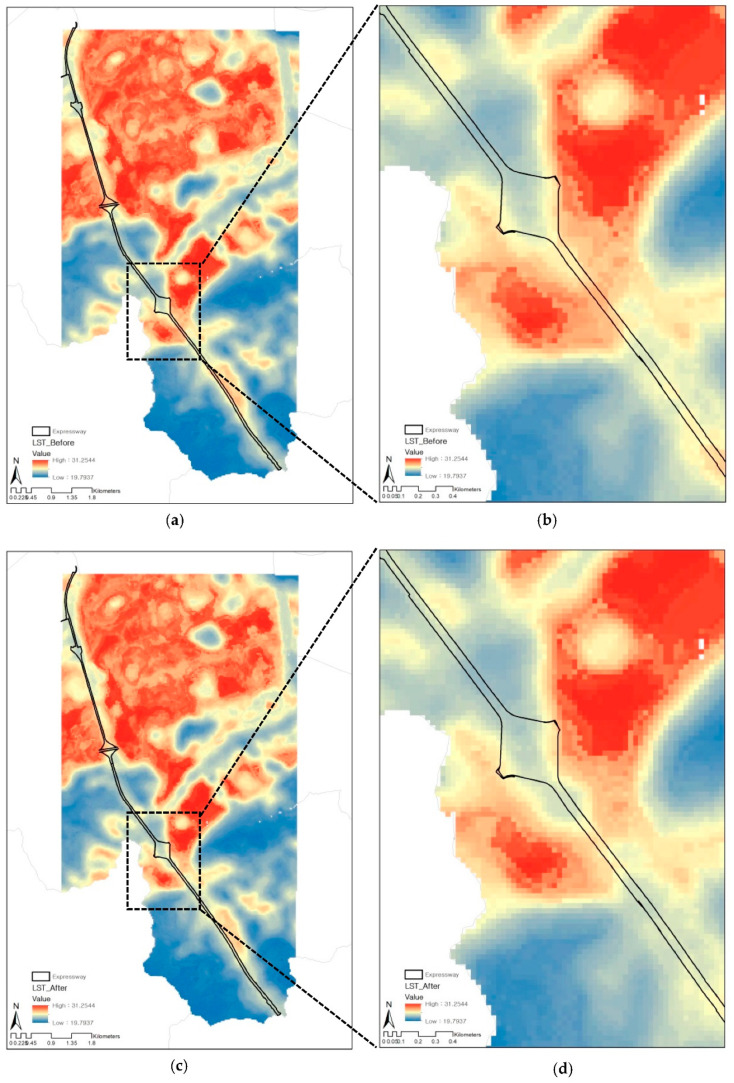
Prediction of LST before (**a**,**b**) and after (**c**,**d**) expressway underground project.

**Figure 9 ijerph-19-15926-f009:**
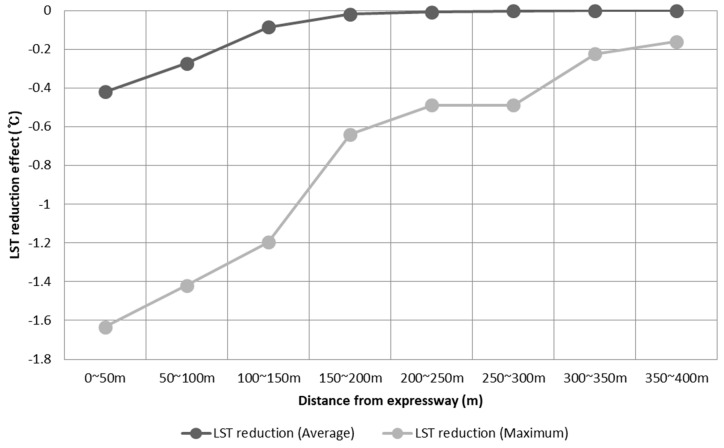
Average and maximum LST reduction after greening project (50 m~500 m).

**Table 1 ijerph-19-15926-t001:** Variables used in the study.

Data			Spatial Resolution	Source (Year)
Dependent Variable	Land Surface Temperature		30 m × 30 m	Landsat 8 (2019)
Independent Variable	Landscape features	NDBI	10 m × 10 m	Sentinel-2 (2019)
		NDWI		
		GNDVI		
	Topographic features	Elevation		Digital Elevation Map (2019)
		Slope		
	LULC features	Urbanized area		Land Cover Map (2019)
		Agricultural area		
		Forest area		
		Grassland area		
		Wetland area		
		Bareland area		
		Water area		

**Table 2 ijerph-19-15926-t002:** Descriptive statistics.

Variables	Mean	Min	Max	SD
LST	27.3514	13.6198	43.1308	4.0636
NDBI	−0.0987	−0.9961	0.8149	0.1847
NDWI	−0.2621	−0.8264	0.4336	0.2284
GNDVI	−0.1715	−0.9598	0.6093	0.2272
Elevation	59.9335	0.1000	752.7495	82.8336
Slope	7.9119	0.0000	69.8180	9.3268

**Table 3 ijerph-19-15926-t003:** Parameters of XGBoost model.

Parameters	Values
Number of iterations	280
Max depth	11
Subsample ratio	0.7
Learning rate	0.2
Colsample_bytree	0.9
Alpha	1 × 10^−6^
Lambda	0.05
Gamma	0

## Data Availability

Not applicable.

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
