# Peer review of "Examining the Relationship between Land Use/Land Cover (LULC) and Land Surface Temperature (LST) Using Explainable Artificial Intelligence (XAI) Models: A Case Study of Seoul, South Korea"

_ijerph, 2022, doi:10.3390/ijerph192315926_

Round 1
Reviewer 1 Report
Overall, methodologically speaking, this study may merit publication, but research-wise, this study lacks a novel research question that addresses an important gap in the literature on urban LST. The only thing catchy is the application of a relatively new AI method to estimating LST based on land cover proportions. The results seem okay, NDVI increasing LST, while GNDVI reducing it. Even though Figure 6 is the key result of the study, it is not presented propertly. It needs a legend, instead of random categorical numbers in the raw data (710, 130, 110...), and deserves another paragraph explaining how to interpret both figures left and right. What do those numbers on the x-axis indicate? When it comes to the scenario testing (3.2.4 LST prediction), provide a figure showing the reference case and the "scenario" (ex. new green space). Also, in Figure 7, it is hard to tell the changes in LST.
Author Response
Overall, methodologically speaking, this study may merit publication, but research-wise, this study lacks a novel research question that addresses an important gap in the literature on urban LST. The only thing catchy is the application of a relatively new AI method to estimating LST based on land cover proportions.
- Thanks for your comments. This study can be differentiated from previous literatures in both methodological and theoretical aspects. In terms of methodological perspectives, this study applied XAI techniques in modeling LST based on surrounding LULC characteristics. For theoretical view, on the other hand, this study is novel in that it developed several LST prediction models with multiple buffer distances (from 50m to 500m) to derive optimal extent of influencing factors on LST. Also, while a majority of previous studies have examined the LST reduction (or increase) effect of certain LULC types (park, river, buildings, etc), this study utilized LST data of whole city to figure out its associations with LULC proportions. We added and highlighted these explanations in the manuscript (p2).
The results seem okay, NDVI increasing LST, while GNDVI reducing it. Even though Figure 6 is the key result of the study, it is not presented properly. It needs a legend, instead of random categorical numbers in the raw data (710, 130, 110...), and deserves another paragraph explaining how to interpret both figures left and right. What do those numbers on the x-axis indicate?
- Thanks for the comments. We unintentionally missed the explanations of each figure and LULC type numbers in Figure 6. We added paragraph explaining each figure regarding SHAP values and replaced the numbers with LULC type name (p11)
When it comes to the scenario testing (3.2.4 LST prediction), provide a figure showing the reference case and the "scenario" (ex. new green space). Also, in Figure 7, it is hard to tell the changes in LST.
- Thanks for the recommendations. First, above figures (a, b) represents the reference case when there’s expressway with no green spaces. On the contrary, below figures (c, d) shows the LST prediction results when expressway is converted to new green spaces. We added these explanations in the manuscripts. Second, we re-ranged the figures so that the changes in LST before and after LULC changes can be highlighted in the figures (p12). Since LST reduction effect was dominant in the vicinity of expressway, we provided zoom-in figures for before and after the greening project.
Reviewer 2 Report
Comments on [IJERPH] Manuscript ID: ijerph-2021213
By integrating XGBoost and SHAP models, this paper establishes a prediction model of surface temperature change in Seoul using several different buffer distances. It has certain practical guidance for the local government to improve the urban heat island effect and control the thermal environment. The framework structure of the article is relatively reasonable. Overall, the paper is structured in a solid way, with a detailed description of the research context, methodologies, and results. As a reviewer, however, my problem with this paper rests in the clarity and validity of the methodologies and terms used.
1. What scientific problem is addressed in this passage? What is its contribution?
2. The abstract lacks the explanation of important influencing factors, and the front and back logic needs to be strengthened;
3. In the "2.1. Study area" section, there are relatively few descriptions of the reasons why Seoul was chosen as the study area. It is necessary to show its uniqueness, and what contribution it has made in this study area;
4. Line 140 "Table 1. Description of variables used in the study" lacks the explanation and description of the selected indicators. It is suggested that certain explanation and elaboration be carried out to make it more targeted and scientific.
5. It is recommended to carefully check the language description in the text. For example, there are obvious errors in line 221 'The area and proportion of each LULC type in Seoul are described in Table 4.' but the actual expression should be 'Table 3';
6. It is suggested that the explanation on NDBI and GNDVI in line 316 be moved forward to '2. Materials and Methods'.
7. Several policies on urban thermal environment management proposed by Line 323 need to be described in detail. The current description is only based on the previous research results;
8. The '4. conclusion' part of the article does not give the description of regularity, it lacks the specific description of the contribution to relevant scientific models.
Author Response
By integrating XGBoost and SHAP models, this paper establishes a prediction model of surface temperature change in Seoul using several different buffer distances. It has certain practical guidance for the local government to improve the urban heat island effect and control the thermal environment. The framework structure of the article is relatively reasonable. Overall, the paper is structured in a solid way, with a detailed description of the research context, methodologies, and results. As a reviewer, however, my problem with this paper rests in the clarity and validity of the methodologies and terms used.
- Thanks for your comments. We carefully read all of your comments and revised manuscript as suggested.
- What scientific problem is addressed in this passage? What is its contribution?
- This study proposed several scientific problems in both methodological and theoretical aspects. In terms of methodological perspectives, this study applied XAI techniques in modeling LST to overcome the limitations of both white-box and black-box approaches in previous works. For theoretical view, on the other hand, a majority of previous studies have examined the LST reduction (or increase) effect of certain LULC types (park, river, buildings, etc) rather than LST of a whole area. To fill this research gap, this study utilized LST data of Seoul city to figure out its associations with LULC proportions. Also, this study is novel in that it developed several LST prediction models with multiple buffer distances (from 50m to 500m) to derive optimal extent of influencing factors on LST. Based on our study’s findings, we may contribute to not only assessing thermal and environmental impact of urban planning, but also determining the priorities of different policy measures in improving the thermal environment. These explanations were added and highlighted in the manuscript (p2).
- The abstract lacks the explanation of important influencing factors, and the front and back logic needs to be strengthened;
- Thanks for the comments. We carefully re-read the abstract and strengthened overall logical structure including the explanations and policy implications of study’s findings (p1).
- In the "2.1. Study area" section, there are relatively few descriptions of the reasons why Seoul was chosen as the study area. It is necessary to show its uniqueness, and what contribution it has made in this study area;
- Thanks for the recommendations. Seoul is one of the representative metropolitan areas in worldwide, and has highly mixed land-use patterns throughout the city. Also, LST in Seoul has temporal and spatial variations and thus enable researchers to examine the associations between LST and LULC more variously. We added and highlighted these explanations in the manuscript (p3).
- Line 140 "Table 1. Description of variables used in the study" lacks the explanation and description of the selected indicators. It is suggested that certain explanation and elaboration be carried out to make it more targeted and scientific.
- Thanks for the suggestions. In 2.2.1 and 2.2.2 sections, we fully explained each dependent and independent variable and its description including equations in detail. To avoid misguide, we changed the name of Table 1 since it just summarizes the overall variables (p4).
- It is recommended to carefully check the language description in the text. For example, there are obvious errors in line 221 'The area and proportion of each LULC type in Seoul are described in Table 4.' but the actual expression should be 'Table 3';
- Thanks for the recommendations. We re-read the overall manuscript and check the language description in the text, including the sentence you pointed out.
- It is suggested that the explanation on NDBI and GNDVI in line 316 be moved forward to '2. Materials and Methods'.
- Revised as suggested (p5).
- Several policies on urban thermal environment management proposed by Line 323 need to be described in detail. The current description is only based on the previous research results;
- Thanks for the comments. We supplemented the policy implications on urban thermal environment management. In particular, we added the limitations of current policies in urban and environmental planning and suggested strategies for the improvement based on the findings of our study (p14).
- The '4. conclusion' part of the article does not give the description of regularity, it lacks the specific description of the contribution to relevant scientific models.
- Thanks for the suggestions. We added the description of our study’s regularity and contribution on relevant fields in the ‘conclusion’ part (p14).
Round 2
Reviewer 2 Report
1.The scientific questions and implications can be further deepened;
2. Figure 5 and Figure 8 can be optimized.
3.There are many tables in the article, can you change it to other forms, or change some tables to text narration? For example, Table 2 and Table 3 are linked together, which is not conducive to the presentation of results.
4.There are too many citations, and the ones that are not obvious can be removed.
5.The abstract is a bit long, it is suggested to reduce it and highlight the research significance.
Author Response
- The scientific questions and implications can be further deepened;
- Thanks for your comments. We supplemented the research questions and policy implications more specifically and deeply in the introduction (p2) and conclusion (p14) part.
- Figure 5 and Figure 8 can be optimized.
- Thanks for the comments. We revised suggested figures into more optimized format.
- There are many tables in the article, can you change it to other forms, or change some tables to text narration? For example, Table 2 and Table 3 are linked together, which is not conducive to the presentation of results.
- As you suggested, we converted table 3 into new figures, to increase interpretability of study’s results. Other tables in the manuscript are remained since they seem essential in explaining the overall study.
- There are too many citations, and the ones that are not obvious can be removed.
- Thanks for the suggestions. We checked all citations of the manuscript and removed when they are not highly noteworthy. After the removal, the number of citations in the manuscript changed from 67 to 54.
- The abstract is a bit long, it is suggested to reduce it and highlight the research significance.
- As you suggested, we reduced the overall abstract to highlight the research significance.